# Effect of biologic treatments on growth in children with juvenile idiopathic arthritis: A systematic review

Yasmine Makhlouf [1,2]*, Hiba Ben Ayed[1,2], Saoussen Miladi[1,2], Hiba Boussaa[1,2], Kawther Ben Abdelghani[1,2], Alia Fazaa[1,2], Ahmed Laatar[1,2]

1 Department of rheumatology, Mongi Slim Hospital, La marsa, Tunis, Tunisia, 2 University Tunis El Manar, Tunis, Tunisia

* Yasmine.makhlouf@fmt.utm.tn

## Abstract

Children with Juvenile idiopathic arthritis (JIA) often experience growth retardation due to various factors. The advent of biologic therapies has revolutionized the management of aggressive forms of JIA. This systematic review aims to provide updated insights into the impact of biologic treatments on growth retardation in pediatric JIA patients. Following PRISMA guidelines, we systematically searched Medline, Embase, and the Cochrane Library for eligible articles. Included were cohort studies, trials, and retrospective studies that evaluated growth outcomes in children with JIA receiving biologic therapy. Twelve studies published between 2003 and 2018 were analyzed, encompassing 1513 patients with a mean age of 11.4 years. Tumor necrosis factor alpha inhibitors were the predominant biologic agents used (75.8%), with a mean follow-up duration of 2 years post-biologic therapy initiation. Growth assessment criteria included Height Standard-deviation-score (HSDS), growth velocity, and height velocity (cm/year). Before biologic treatment, 15% of patients exhibited growth delay, while 75.4% had impaired growth. Following biologic therapy, growth delay decreased to 8% and impaired growth to 36.8%. Patients with systemic JIA showed lower changes in growth parameters compared to others, and no significant differences were observed between different biologic drugs. However, lower growth velocity changes were noted in patients treated with multiple biologic agents. Two studies suggested that growth catch-up was most pronounced during the first year of treatment. This systematic review highlights the potential of biologic therapies in mitigating growth impairment associated with JIA. Despite observed positive effects, further research is warranted to elucidate underlying mechanisms and optimize treatment strategies.

## Introduction

Juvenile idiopathic arthritis (JIA) is a chronic autoimmune disease affecting children, characterized by persistent joint inflammation [1]. It is associated with several

**Data availability statement:** All relevant data are within the manuscript and its Supporting Information files.

**Funding:** The author(s) received no specific funding for this work.

**Competing interests:** The authors have declared that no competing interests exist.

complications that can cause short or long-term disability and reduce the quality of life [1]. One of the main complications, is growth retardation which may lead to psychological problems in adolescence and adulthood [2]. In fact, the prevalence of short stature in JIA varies from 10.4% in children with polyarticular disease to 41% of patients with the systemic form [3]. The pathogenesis of growth failure is multifactorial and includes the effects of chronic inflammation, long-term corticosteroid (CS) use, malnutrition, altered body composition, and delayed onset of puberty. [4–6]. Moreover, the degree and duration of disease activity are also important, as these factors may have systemic effects on the GH-IGF-1 axis or local effects on growth plate homeostasis and function [7–9]. Advances in drug treatment, especially anti-tumour necrosis factor (anti-TNF □) and other inflammatory pathways have provided new means of controlling the most aggressive forms of JIA [10–12]. In fact, biological therapy can enduce rapid remission and faster CS reduction [10,12]. While treatment strategies have evolved over the years, the impact of these treatments on growth in pediatric patients remains an area of interest and concern. In this context, some authors have discussed the potential effect of biologics on growth velocity throught the extinction of the disease activity [13–15]. Understanding the effect of biologic therapies on growth parameters is crucial for optimizing treatment strategies and improving long-term outcomes in children with JIA.

This systematic review aimed to provide the most up-to-date information about the effect of biologic treatments on growth in children with JIA.

## Materials and methods

All data analyzed were extracted from published studies. For the present paper, no ethical approval or written informed consent was required. The search strategy, literature selection, and data extraction were conducted by two investigators (Dr YM and Dr BAH) independently, then discussed, and any disagreement was resolved by consensus.

This systematic review was registred in prospero: CRD42023432733.

### Search strategy

This systematic review followed the preferred reporting items for systematic reviews guidelines (PRISMA) [16]. Eligible articles were searched in Pubmed, Scopus, and Cochrane Library. For PubMed, the search was carried out using a strategy employing the combination of the MeSH (Medical Subject Headings) terms associating the combination of synonyms of "juvenile idiopathic arthritis", terms related to "growth", and terms related to "biological therapy" (Table 1). For Scopus and Cochrane Library, the previous terms were searched in the article title, abstract, or keywords. In addition, the reference lists of the included articles were checked. The first authors of this systematic review (Dr MY and Dr BAH) agreed on the articles to be included in this paper. All the aspects of systematic review methods were specified before starting the review.

### Selection criteria

Inclusion criteria: A comprehensive search was conducted from the 1st January 2000 until October 15th 2023. Only full-text papers available and written in English

**Table 1. Characteristics of JIA patients.**

| | N | Mean age in years at disease onset, extremes | Mean Disease duration, extremes | Mean actual age in years, SD, extremes | Mean age in years at biologic initiation | Percentage of patients with active disease (%) |
|---|---|---|---|---|---|---|
| Schmeling et al | 7 | 3.2 [0.5-7] | 7.8 [5- 16.5] | NM | NM | 100% |
| Tynjala P et al | 53 | 3.7 [0.8-13.3] | 5.8 [0.3-13.7] | 3.7 | 9.6 [3-14.7] | 48% |
| Fernandez V et al | 31 | 4.3 [1–9] | NM | NM | 9.6 [2.1-13.7] | NM |
| Biliau et al | 16 | NM | 2.2 [0.3- 9] | 9 [3.8-15.5] | NM | 100% |
| Giannini et al | 594 | 58.1±44.5* | | NM | NM | NM |
| Uettwiller F et al | 100 | 4.3 [0.7-13.7] | 1.8 [0.18- 11.1] | NM | 7.1 [1.5-15.6] | 45% |
| Miyamae et al | 45 | NM | 3.4 [1-7.1] | 8.1 (4.2) | NM | NM |
| Shafferman et al | 167 | 7.9 [4.0–12.7] | NM | 11 (10.2) [6.6–14.2] | NM | 26% |
| Świdrowska et al | 21 | 8.5 [3–15] | 5.5 [3–12] | NM | 10.6 [5–15] | NM |
| Kearsley-Fleet et al | 191 | NM | 3.5 [1.7–7.1] γ | 11[7.3–12.9]γ | NM | 63.8% |
| De Benedetti et al | 83 | NM | 4.9±4.1 | 9.2±4.2 | NM | 100% |
| Bharucha et al | 187 | NM | 4.2±3.7 | 11.0 | 11.0±4.0 | 100% |

*N: number of patients, NM: Not mentioned; JIA: Juvenile idiopathic arthritis; DMARDs: disease-modifying-antirheumatic drugs; ± standard deviation, γmedian;*

*mean age at disease onset in the biologic group.

were considered and were required to meet the following criteria: (1) publication in peer-reviewed journals, (2) relevance to the research question, and (3) availability of sufficient data for extraction in addition to your other already included study specific criteria. We selected original articles (observational, randomized trials) in this systematic review. Studies were included if they assessed: 1) Patients diagnosed with JIA according to the ILAR (International League Against Rheumatism) 2001 classification criteria (1), and treated with biologics. 2) JIA children who who either have initial impaired or normal growth or growth promotion in response to biologic treatments. 3) Studies evaluating clinical, biological, and radiological growth parameters before and after biological therapy. 4) The study was approved by the local Ethics Board and informed consent of parent's children was obtained. Additional articles were manually retrieved based on the references of selected articles. If any study included overlapping data, the most comprehensive one was selected.

Exclusion criteria: 1) Patients with JIA associated to other autoimmune (juvenile systemic lupus erythematosus) diseases that may compromise the growth velocity, unless they were blended into the studied population and we could not differentiate them when interpreting results. 2) Papers written in another language than English. 3) Publications not representing original research (i.e.,; reviews, editorials, qualitative papers, case reports, and letters to editors) were excluded. Additionally, articles that did not align with the objectives of the systematic review were excluded.

## Data extraction and quality assessment

Extracted data from each study was scrutinized independently by both investigators (Dr MY and Dr BAH). They used a pilot-tested extraction form. In cases where consensus could not be reached, a third party (Dr BH) was consulted to resolve disagreements. Missing data were handled by first attempting to contact study authors for additional information.

The extracted data included the main methodological characteristics of the articles, such as the year of publication, country, study design, number of subjects, mean age of subjects, inclusion and exclusion criteria, and duration of follow-up. Our main judgement criteria was the improvement of the growth parameters before and after biologic treatment. Furthermore, we identified potential biases using Newcastle Ottawa Scale (NOS) for the cohort and case control studies and the revised Cochrane risk-of-bias tool for randomized trials (RoB 2) [17,18].

## Results

The systematic review encompassed 12 selected papers, chosen from an initial pool of 97 papers identified through a search process outlined in Fig 1.

### Characteristics of the studies

The main characteristics of the 12 studies retained in this systematic review are represented in S1 Table.

The studies, published between 2003 and 2018 [19,27], were conducted across various countries, including Germany [19], Finland [15], Sweden [20], Belgium [21], Ohio [13], France [22], Japan [14], the United States [23], Poland [24] and the United Kingdom [25].

Notably, two studies [26,27], the TENDER trial and the CHERISH trial, were multicenter trials conducted by members of the Paediatric Rheumatology International Trials Organisation (PRINTO) [28] and the Pediatric Rheumatology Collaborative Study Group (PRCSG) [29], respectively."

The applied designs were the cohort study [14,15,19–22,25], the open-label non randomized trial [13], the open-label randomized placebo-controlled trial [26,27], the case control study [23] and the retrospective study [24].

Exclusion criteria were explicitly stated in almost all studies (11 out of 12), with common criteria including previous growth hormone (GH) therapy [15,19,20,22,24,26,27], age over 15 years at the initiation of biologic treatments [15], advanced puberty [20], postpubertal patients at biologic therapy onset [22], major concurrent medical conditions [13,21,22], and biopsy diagnosis of inflammatory bowel disease [23].

### Characteristics of the patients

The main characteristics of the patients are represented in Table 1.

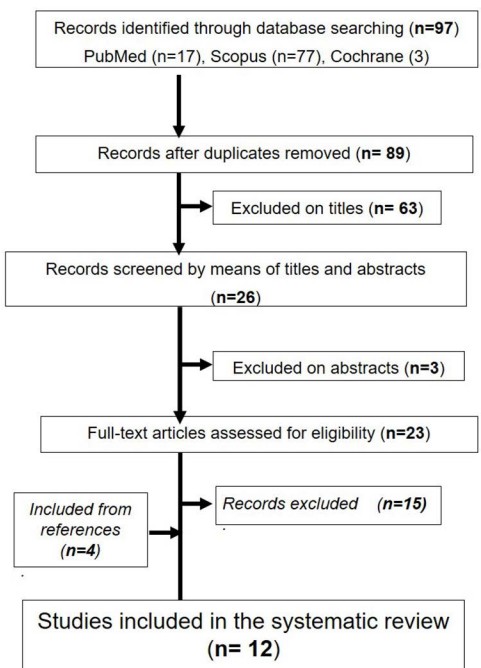

**Fig 1. Flowchart outlining the protocol adopted in this systematic review.**

A total of 1513 patients were included, with sample sizes ranging from 7 to 594 across studies [13,20]. When reported, the mean age of the patients was 11.4 years, with extremes ranging from 6.4 to 11 years [15,19,27]. The mean age at disease onset ranged from 3.2 to 8.5 years, with a mean of 5.3 years [19,24]. The mean age at initiation of biological treatment ranged from 7.1 [22] to 11 years, with a mean of 9.7 years [22,25,27].

The mean duration from the diagnosis of JIA to enrolment was mentioned in 10 studies and ranged from 1.8 to 7.8 years, with a mean of 4.3 years [19,22]. All studies applied the 2001 International League of Associations for Rheumatology (ILAR) criteria for JIA diagnosis. The distribution of JIA subtypes was as follows: polyarticular (n = 603), systemic (n = 283), oligoarticular (n = 145), enthesitis-related arthritis (n = 67), psoriatic arthritis (n = 35). Some patients were no categorized (n = 17) [13–15,19–26], while in one study, all patients (n = 187) had polyarticular or extended oligoarticular JIA [27].

## Characteristics of treatments modalities

The main reported treatments before biologic therapy onset were conventional Disease-modifying-antirheumatic drugs (DMARDs) (Methotrexate (n = 567) [13,19,23,24], Sulfazalsine (n = 8) [19,24], Hydroxychloroquine (n = 2) [24], Azathioprine (n = 1) [19]), non-steroidal anti-inflammatory drugs (NSAIDS) (n = 83) [26], intravenous immunoglobulin (n = 4) [15,19], and cyclosporine (n = 2) [20]. The mean CS dose (n = 454) before biologic therapy was 0.4 mg/kg/day [0.1–2.9] [14,15,19,21,23,24,26,27]. Regarding biologic use, anti-TNF □ was the most used biologic: Etanercept (n = 6), Infliximab (n = 2), Adalimumab (n = 1), Golimumab (n = 1. The anti-TNF□ used was not precised in two studies. Tocilizumab was used in 4 studies [14,24,26,27]. Only 47 patients received two biologics concurrently [22,24]. Anti-IL1 and Abatacept were used one study each. Mean follow-up duration after biologic therapy was 2.1 years with extremes ranging from 0.5 [22] to 3 years [13,14,19].

## Growth assessment criteria

This summary outlines the various parameters used for growth assessment in the studies, including anthropometric data, bone health indicators, and blood markers related to growth and bone metabolism. Growth assessment criteria included anthropometric data, growth velocity, bone health, and blood markers. Anthropometric data involved measurements such as height, weight, and body mass index (BMI). Height was expressed in centimeters (cm) [13,17], percentiles [13], height standard deviation score (HSDS) [14,15,19,21,22,24,26,27], adjusted for weight (%) [15], using Tanner's formula [22,32], or adjusted parental HSDS [20]. Height velocity was also measured, expressed in centimeters per year [19,25–27] or as a standard deviation score (SDS) [14]. Weight was recorded in kilograms (kg) [13] and as percentiles [13], while BMI was expressed in kg/m² [13,15,23], percentiles [13], or SDS [23,25,26]. Growth velocity was assessed using the change in HSDS (Δ HSDS) during follow-up [14,15,20–22,25]. Bone health parameters included bone mineral density (BMD) and bone mineral content [21], and the rate of bone maturation [15]. Blood markers used for assessment were insulin-like growth factor-1 (IGF-I) serum levels [19,26], insulin-like growth factor binding protein-3 (IGFBP-3) serum levels [19], and serum levels of soluble markers of osteoblast/osteoclast activity [26].

## Growth assessment parameters after biologic therapy

Across the studies, significant changes in growth parameters were observed following biologic therapy, although the degree of improvement varied (Table 2). Before biologic treatment, 15% of patients exhibited growth delay, while 75.4% had impaired growth. Following biologic therapy, growth delay decreased to 8%, and impaired growth was reduced to 36.8%. Significant decreases in HSDS were reported in four studies [19,25–27], while a statistically significant increase in height velocity was observed in four studies [14,15,19,26]. An increase in growth velocity was noted in children with delayed growth compared to those with normal growth [15,21], as well as in all patients in one study [26]. Biologic markers also showed significant changes, with two studies reporting a significant increase in IGF-I serum levels [19,121] and one study noting a significant increase in serum levels of soluble markers of osteoblast/osteoclast activity [26]. In terms of bone health, lumbar BMD significantly increased after biologic therapy in one study [23], although there was no

**Table 2. Growth assessment parameters before and after biological treatment in the different studies.**

| | Growth parameters | | Before biological treatment | After biological treatment |
|---|---|---|---|---|
| **HSDS** | **Mean** | Schmeling et al. 2003 | -2.4 [-3.9, -1] | -1.9 (after 1 year)* <br> -1.1 (after 2 years)* |
| | | Fernandez Vojvodich, et al. 2007 | -2.1 [-3.8, -0.5] | -1.9 (after 1 year) |
| | | | -1.3 [-2.6, 0.9] | -1.1 (after 1 year) |
| | | Biliau et al. 2010 | -0.6 [-4.3, 1.6] | NS |
| | | Uettwiller F et al. 2013 | −0.3 [−3.6, 2.9] | -0.15 [−4.9, 3.5] (NS) |
| | | | -0.3 [-4.5, 2.3]$ | -0.35 [-6, 2.7] (NS) |
| | | Miyamae et al. 2013 | -2.7 ± 1.9 | NM |
| | | Świdrowska et al. 2015 | 0.4 ± 1 | −0.1 ± 1.1 |
| | | Kearsley-Fleet et al. 2015 | -0.7 (±1.4) | -0.5 (±1.4) after 1 year <br> -0.5 (±1.4) at 2 years |
| | | De Benedetti et al. 2015 | -2.2 ± 1.9 | -1.9*** (after 1 year) <br> -1.9*** (after 2 years) |
| | | Bharucha et al. 2018 | −0.5 ± 1.2 | -0.2 (after 2 years) |
| | **(HSDS < - 2), (n)** | Uettwiller F et al. 2013 | 15 patients | 10 patients |
| | | Kearsley-Fleet et al. 2015 | 29 | 13 |
| | | Bharucha et al. 2018 | 22 | NM |
| **Other means of HSDS evaluation** | Fernandez Vojvodich, et al. 2007 | **ΔHSDS in all patients** | -0.25 | 0.2 (after 1 year) |
| | | **ΔHSDS in the prepupertal group** | -0.3 ± 0.1 | 0.2 ± 0.1*** (after 1 year) |
| | | **ΔHSDS in the prepupertal group** | -0.2 ± 0.2 | 0.2 ± 0.1 (NS) |
| | | **Low ΔHSDS (<-0.5)** | 43 patients | 18 patients |
| | | **Impaired ΔHSDS (<0)** | 76 patients | 46 patients |
| | Tynjala P et al. 2006 | **ΔHSDS<0** | 53 patients | 17 patients |
| | | **ΔHSDS ↓ (Mean)** | -0.35 [-0.5, -0,28] | 0.1 [0.5, 0.12]*** |
| | | **ΔHSDS in patients with previously normal growth (Mean)** | 0.1 [0, 0.18] | 0.15 [0, 0.2] (NS) |
| | Biliau et al. 2010 | **ΔHSDS** | NM | Significant increase after 12 * |
| **Height velocity (cm/year)** | Schmeling et al. 2003 | | 3.6 ± 1.2 | 7.6 ± 1.2 (first year)*** <br> 6.2 ± 1.4 (second year)** |
| | Tynjala P et al. 2006 | | NM↓ | Increased by +1.8 *** |
| | | | NM | NS |
| | Miyamae et al. 2013 | | -5.9 (after 1 year) | -2.5** (after 1 year) |
| | Kearsley-Fleet et al. 2015 | | NM | 5.8 |
| | De Benedetti et al. 2015 | | 3.0 ± 2.5 | 6.6 (after1 year)*** <br> 6.8 (after 2 years)***. |

*(Continued)*

**Table 2.** (Continued)

| | Growth parameters | | Before biological treatment | After biological treatment |
|---|---|---|---|---|
| Anthropometric data | Tynjala P et al. 2006 | Mean height adjusted relative weight ↓ | 11.2% [4.9- 19.4] | 15.4% [3.1- 24]* |
| | | Mean height adjusted relative weight in patients with previously normal growth | NM | NS |
| | | Mean BMI (kg/m²)↓ | 18.5 [17.3-19.6] | 19.9 [18.5-21.3]*** |
| | | Mean BMI in patients with previously normal growth (kg/m²) | 18 [16.4-19.7] | 19.2 [17.2-21.2]* |
| | Giannini et al. 2010 | Mean height (cm)/ (percentile) | 139.2/ 42 | NM/42.7 (third year) * |
| | | Mean weight (kg)/ (percentile) | 41.4/ 51.6 | NM/55.7 (third year) (NS) |
| | | Mean BMI (kg/m²)/(Percentile) | 19.8/57 | NM/61.8 (NS) |
| | Miyamae et al. 2013 | Mean BMI (SDS)/ (kg/m2) | 0.5 (1.21)/ 20.1 | 0.6*/21.9* |
| | | BMI category (n): obese/overweight | 24/28 | 33/30 (NS) |
| | Shafferman et al. 2014 | Mean weight (kg) | 37 [22–55] | NM |
| | | Mean height (cm) | 140 [119–162] | NM |
| | Kearsley-Fleet et al. 2015 | Mean BMI, (SDS) | 0.7 | NM |
| | De Benedetti et al. 2015 | Mean BMI, (SDS) | 0.8 ± 1.5 [0.7] | No significant change over the time |
| Biological markers | Schmeling et al. 2003 | Mean IGF-I/ Mean IGFBP-3 serum level (ng/mL) | 177±62/5.2±0.6 | 432±193 ***/5.9±0.5*** |
| | De Benedetti et al. 2015 | Mean serum level of IGF-1 (SDS) | −1.0±1.4 | -0.2 (after 1 year)*** -0.1 (after 2 years)*** |
| | | Mean serum level of OC | NM | Increased significantly (after 2 years) *** |
| | | Mean serum level of CTX-1 | NM | Increased significantly (after 2 years) *** |
| | | The mean (SD) ratio of OC to CTX-1 | 54 | 66 (after 2 years)* |
| Densitometric data | Biliau et al. 2010 | BMC (%) | 2.6 [1.8, 3.3] | 3.2 [2.4–3.5]}* (after 1 year) |
| | | Lumbar BMD, z score | -1.4 [-3.2, -0.1] | -1 [-3, -0.4]* (after 1 year) |

*SDS: Standard-deviation-score; TNF: Tumor-necrosis-factor; CID: chronic inflammatory diseases; GH: Growth hormone; IGF-1: insulin-like growth factor-1; IGFBP-3: insulin-like growth factor (IGF)-binding protein-3; BMI=Body Mass Index; BMD: Bone minearl density; BMC: Bone mineral content; OC: a soluble marker of osteoblast activity; CTX-1: a soluble marker of osteoclast activity, *p ≤ 0.05; **p ≤ 0.01; **p ≤ 0.001; SD: Standard-deviation; NM: Not mentioned; NS: Not significant; S: Significant;HSDS: Height SDS defined as observed height minus mean height for age divided by SD, where SD was the standard deviation for the normal population of the same chronological age and sex; +: ΔHSDS≥ 0: catch up growth and ΔHSDS<0: impaired growth; height adjusted relative weight (%): the ratio of weight for height (W/H) in (kg/cm) to the mean W/H in the normal population of the same calendar age and sex; $ adjusted for height; γ at Tanner stage ≤ 3 at baseline; ↓delayed growth.*

statistically significant improvement in bone maturation in another study [15]. Additionally, significant improvement in BMI was reported in three studies [14,15,23].

## Factors interfering with growth outcomes after biologic therapy

Growth outcomes varied with puberty stage. An increase in IGF-1 levels was observed in both prepubertal and pubertal children [19], but while the majority of prepubertal and pubertal patients experienced growth improvement after treatment with Etanercept, the growth improvement in the pubertal group did not reach statistical significance [20]. Furthermore,

significant increases in mean HSDS after two years of TCZ treatment were particularly noted in patients with Tanner stage ≤ 3 at baseline (72%) [27]. Patients with systemic JIA had significantly lower changes in growth parameters on biologic treatment compared to other patients in some studies [22,25]. An inverse correlation between IGF-1 serum levels and CRP levels was demonstrated in some studies [19,26], and improvement in disease activity was associated with an increase in height velocity and growth catch-up during the first year of TCZ treatment [26]. Etanercept effects on growth were associated with a reduction of systemic IL-6 [21]. However, no consistent effect of disease control on height was observed in other studies [13,22,26]. The duration of the disease also impacted growth outcomes, with an inverse correlation between baseline HSDS and disease duration noted in one study [14] but not in another [27]. Previous corticosteroid exposure and/or a high dose of corticosteroids were significantly associated with lower changes in growth velocity after biologic treatment in some studies [14,25,27], but these results were not verified in another study [22]. Lastly, significant growth catch-up occurred during biological treatment, particularly among patients with delayed growth in some studies [15,27], while growth retardation persisted in children with pre-existing growth retardation in another study [22]. Overall, the most pronounced growth catch-up occurred within the initial six months of treatment [24], with significant growth catch-up observed during the first year of treatment [19].

## Discussion

### Main findings

This systematic review provides comprehensive insights into the effects of biologic therapy on growth outcomes in children with JIA. The analysis of 12 selected studies revealed an overall significant improvement in growth parameters following biologic therapies, such as Etanercept and Tocilizumab. Specifically, the review highlights increased growth velocity, HSDS particularly in patients with delayed growth at baseline. These improvements were most notable in the first year of treatment.

### Implications of these findings

Several growth parameters were used across studies using different metrics such as IGF-1 levels, SDS, and height, which capture distinct aspects of growth. For example, IGF-1 levels primarily capture growth disturbances related to hormonal dysregulation, while SDS and height provide anthropometric measures, making them complementary but not interchangeable.

The findings of this systematic review suggest that biologic therapies could be an essential component of the management strategy for growth retardation in children with JIA, offering benefits beyond disease control by potentially enhancing growth. Specifically, initiating biologic therapy early in the disease course, is beneficial for patients presenting with growth delays or high disease activity as early control of systemic inflammation can mitigate its negative effects on growth velocity and overall development.

Growth retardation is a great concern in JIA patients. In fact, the Childhood Arthritis Prospective Study (CAPS) showed that 39% of JIA patients experienced growth restriction (defined as Δ HSDS < 0.5) over the first 3 years of disease [4]. According to the results of the present review, impaired growth (Δ HSDS <0) was found in 15% of the patients. The pathogenesis of growth disorders is multifactorial and the role of chronic inflammation was discussed in several studies [2,3,7,30–33]. Pro-inflammatory cytokines play a crucial role in the GH–IGF axis in JIA patients [34]. Beyond its role in inflammation, TNF exerts direct effects on growth and development. Indeed, it can modulate growth hormone secretion, inhibit chondrocyte proliferation and differentiation, and disrupt the growth plate architecture, thereby impairing longitudinal bone growth [34,35]. In conditions of chronic inflammation in RD, specifically JIA, elevated TNF levels can disrupt the normal growth hormone axis and inhibit IGF-1, a key mediator of growth hormone effects. Moreover, TNF-induced alterations in bone metabolism and turnover can affect bone mineralization and contribute to decreased bone density and increased

fracture risk in children [36]. By targeting TNF activity and suppressing the chronic inflammation, TNF inhibitors help mitigate these detrimental effects on growth and bone health, although the precise mechanisms underlying the growth-promoting effects remain to be fully elucidated [37]. In addition to their effect in inhibiting the activity of pro-inflammatiory cytokines, biologics cause rapid remission and are thus CS sparing [14].

However, it's important to note that there are differences in growth outcomes based on factors such as puberty stage with some studies showing a less significant growth improvement in pubertal patients compared to prepubertal ones, and higher disease activity levels exhibited lower changes in growth parameters compared to patients with lower disease activity. In addition, corticosteroid exposure, the timing and duration of treatment may influence the magnitude of growth improvement. Some others suggest that the effect of these drugs is mainly indirect, rather than a direct effect on growth or skeletal maturation. Indeed, IL-6–transgenic mice have growth plates of reduced width, severe alterations in cortical and trabecular bone microarchitecture and impaired maturation of epiphyseal ossification nuclei [38]. These findings might explain an important fact constated in previous studies, is that systemic JIA, a subtype closely related to IL-6, is associated with higher risk of growth failure [3,30]. Systemic JIA patients had significantly lower changes in the growth parameters after anti-IL6 treatment than other patients [22,25]. This also might be explained by the growth inhibitory effects of CS commonly prescribed in this JIA subtype. In fact, the study of Kearsley-Fleet et al. showed that no oral CS use at baseline was associated with improvement in HSDS. Still, according to the authors, this might be a marker of baseline disease severity [25].

### Limitations of the study

To the best of our knowledge, this is the first systematic literature review focusing on the effect of biologic treatments on growth in children with JIA. Additionally, the review highlights that the effects of biologic therapies on growth can vary based on factors such as puberty stage, disease activity levels, corticosteroid exposure, and the timing and duration of treatment.

As another highlight of our review, we focused on the different factors interfering with growth. However, some limitations should be addressed. Firstly, many of the studies included in the review had an observational, non-randomized design with a limited number of patients [19,21,24], which may affect the generalizability of the results. Secondly, there was considerable heterogeneity in the criteria used to assess growth across studies, making comparisons and interpretations challenging.

One main advantage is the robust follow-up for most of the studies. Hopefully, future trials should tackle these particular issues and studies should include more cases and control subjects to ascertain the specific implication of each subset for a better holistic approach.

Studies with a more robust design (RCTs) are a key step toward minimizing bias and improving the quality of evidence. Specifically, RCTs could standardize inclusion criteria, define consistent growth assessment parameters (e.g., SDS, growth velocity, or IGF-1 levels), and control for confounding factors such as disease severity and prior treatments.

### Conclusion

The evidence gathered from this systematic review highlights the potential of biologic therapy in addressing growth retardation in JIA patients. By effectively addressing inflammation, biologics are promising intreating growth retardation, particularly when administered early in the disease course. However, further research is necessary to optimize treatment protocols and elucidate the long-term effects of biologic therapy on growth. Indeed, there is a critical need for further studies on their long-term effects specifically addressing which biologic is associated with the most favorable growth outcomes. Recommendations for early intervention and regular growth monitoring could enhance clinical applications. Moving forward, a holistic approach that integrates both anti-inflammatory and growth-promoting strategies is crucial for enhancing the overall health and well-being of children with JIA.

## Supporting information

**S1 Table. Main characteristics and results of the selected studies aiming to assess the effect of biologics on growth in juvenile idiopathic arthritis patients.**
(DOCX)

**S2 Table. Risk of biais assessment.**
(DOCX)

**S3 Table. Prisma checklist.**
(DOCX)

**S4 Table. List of excluded articles.**
(XLS)

## Acknowledgments

NONE

## Author contributions

**Conceptualization:** Yasmine Makhlouf, Hiba Ben Ayed, Alia Fazaa, Ahmed Laatar.

**Data curation:** Yasmine Makhlouf.

**Investigation:** Yasmine Makhlouf, Alia Fazaa.

**Methodology:** Yasmine Makhlouf, Hiba Ben Ayed, Alia Fazaa.

**Resources:** Yasmine Makhlouf, Hiba Boussaa.

**Software:** kawther Ben Abdelghani.

**Supervision:** Saoussen Miladi, kawther Ben Abdelghani, Ahmed Laatar.

**Validation:** Yasmine Makhlouf, Hiba Ben Ayed, Saoussen Miladi, Hiba Boussaa, kawther Ben Abdelghani, Ahmed Laatar.

**Visualization:** Yasmine Makhlouf, Saoussen Miladi, Hiba Boussaa, kawther Ben Abdelghani, Alia Fazaa, Ahmed Laatar.

**Writing – original draft:** Hiba Ben Ayed.

**Writing – review & editing:** Yasmine Makhlouf.

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
