## [Editor Report · Decision Letter 0]

8 Aug 2024

PONE-D-24-30426Effect of biologic treatments on growth in children with juvenile idiopathic arthritis: a systematic reviewPLOS ONE

Dear Dr. Makhlouf,

Thank you for submitting your manuscript to PLOS ONE. After careful consideration, we feel that it has merit but does not fully meet PLOS ONE’s publication criteria as it currently stands. Therefore, we invite you to submit a revised version of the manuscript that addresses the points raised during the review process.

We look forward to receiving your revised manuscript.

Kind regards,

Anna Denee’ Ware, MPH, MS

Academic Editor

PLOS ONE

Journal Requirements:

"NO authors have competing interests"

3. Please include a caption for figure 1. 

Additional Editor Comments:

This systematic review provides updated insights into the impact of biologic treatments on growth in pediatric patients with juvenile idiopathic arthritis (JIA). While an addition to scientific literature, I believe that a major revision of the write up is warranted. Please see my comments below:

- Please keep all percentage points referenced to the same decimal points for consistency. For example, in the introduction, authors write “10.4% in children with polyarticular disease to 41% of patients with the system form.”

- In the introduction, is the main focus on the effects of biologic treatments for growth “retardation” in children with JIA? I think this statement could use a little clarification for the reader.

- In the first paragraph of the methods section, can you add more detail to how disputes were resolved? In cases where consensus could not be reached, was a third party consulted to arbitrate the decision?

- Instead of stating “The two first authors of this systematic review agreed on the articles to be included in this paper” I would advise to specify the two authors involved as you did in the first paragraph of the methods section.

- For the inclusion criteria, what is meant by “full papers”? Does this mean “full-text papers?” To ensure a robust selection process, please review that studies included were required to meet the following criteria: (1) publication in peer-reviewed journals, (2) relevance to the research question, and (3) availability of sufficient data for extraction in addition to your other already included study specific criteria.

- Please also include the date of inception in the first sentence for clarity.

- In the inclusion criteria section, please review the numbered points on studies selected, they skip number 3.

- In the inclusion criteria section, this sentence “Studies assessing clinical and/or biological and/or radiological growth parameters before and after biological therapy” could be revised for better clarity.

- For number 1 of the exclusion criteria, please add an example of “other autoimmune diseases that may compromise the growth velocity”.

- Please revise the last sentence of the exclusion criteria to improve clarity and maintain a formal tone. Specifically, change "Besides" to "Additionally" and rephrase "not in compliance with the systematic review purpose" to "that did not align with the objectives of the systematic review."

- In the Data Extraction and Quality Assessment section, please revise the following sentence “The extracted data included the main methodological characteristics of the articles: study data (year of publication, country, study design, number of included subjects, mean age of included subjects, inclusion and exclusion criteria, duration of the followup)” for better clarity and readability. Specifically, change "included the main methodological characteristics of the articles: study data (year of publication, country, study design, number of included subjects, mean age of included subjects, inclusion and exclusion criteria, duration of the followup)" to "included the main methodological characteristics of the articles, such as the year of publication, country, study design, number of subjects, mean age of subjects, inclusion and exclusion criteria, and duration of follow-up."

- In the Data Extraction and Quality Assessment section, please fix the spelling of “judgement”

- In the Data Extraction and Quality Assessment section, what does “high quality” mean? Please expand on this.

- In the results, this statement: “The studies, published between 2003 (17) and 2018 (23),” makes it seem as though there were 17 studies selected in 2003, and 23 studies selected in 2018. Since you state “between”, this is not clear. Please revise for clarity and readability.

- In the results section, “The” does not have to be capitalized before “United States” or “United Kingdom”

- For this statement “Notably, two studies (24,25) were multicenter trials conducted by the members of the Paediatric Rheumatology International Trials Organisation (PRINTO) (24) and the Pediatric Rheumatology Collaborative Study Group (PRCSG) (27): The TENDER trial and the CHERISH trial (28, 29) . The applied designs were the cohort study (13,17–20,23), the open-label non randomized trial (11), the open-label randomized placebo-controlled trial (12,24,25), the case control study (21) and the retrospective study (22)”, I am getting confused on which studies were the “two notable studies.” Please revise the statement for better clarity and to clearly identify the references for the notable studies and study designs. For example, change "Notably, two studies (24,25) were multicenter trials conducted by the members of the Paediatric Rheumatology International Trials Organisation (PRINTO) (24) and the Pediatric Rheumatology Collaborative Study Group (PRCSG) (27): The TENDER trial and the CHERISH trial (28, 29)" to "Notably, two studies, the TENDER trial and the CHERISH trial, were multicenter trials conducted by members of the Paediatric Rheumatology International Trials Organisation (PRINTO) (24) and the Pediatric Rheumatology Collaborative Study Group (PRCSG) (27), respectively."

- For the results section, when reporting on patient characteristics, the placing of the references is confusing. Please revise for readability.

- Table 1 could be improved for better clarity and readability. Ensure consistent formatting and alignment of columns and rows, and use clear, concise column headers without abbreviations. Separate different types of data into subsections or separate tables, and include footnotes to explain any abbreviations or unique terms. Highlight key data points with bold or italicized text, and add summary statistics like means and medians. Use visual aids to distinguish groups or highlight differences, and maintain consistency in units of measurement throughout the table. Remove redundant information and provide a clear caption summarizing the table’s content along with any relevant notes.

- In the results 1.4 Growth Assessment Criteria, this section would read easier in paragraph format for a scientific journal. This follows into sections 1.5 and 1.6, which read more like a paper outline instead of a finished manuscript.

- Please add a review of all biologics that were included in the studies reviewed prior to the discussion section.

- The discussion section provides a comprehensive summary of the findings, emphasizing the positive impact of biologic therapies on growth in children with JIA. However, it would benefit from more structured presentation and clearer emphasis on key points.

- Consider organizing the discussion into distinct paragraphs that separately address the main findings, the implications of these findings, and the limitations of the study. Highlight the most significant improvements in growth parameters, such as HSDS and growth velocity, and discuss their clinical relevance.

- In the discussion section - while the potential mechanisms and differences based on factors like puberty stage and JIA subtype are well-covered, it would be helpful to further elaborate on the practical implications for treatment strategies and patient management.

- Lastly, explicitly state the limitations of the review, such as heterogeneity in growth assessment criteria and the observational nature of some studies, to provide a balanced perspective.

---

## [Author Response · Author response to Decision Letter 1]

30 Aug 2024

Dear reviewers, thank you for your pertinent remarks. We hope that we have adequately responded to all your comments.

Journal comments

The manuscript was modified according to plos one requirements.

"NO authors have competing interests"

The statement was added as requested

2. Please include a caption for figure 1.

A caption was added for figure 1

3. Please include captions for your Supporting Information files at the end of your manuscript, and update any in-text citations to match accordingly.

Captions for supporting informations were added at the end of the manuscript.

Additional Editor Comments:

All the revision was marked in done in revision mode and highlighted in green.

This systematic review provides updated insights into the impact of biologic treatments on growth in pediatric patients with juvenile idiopathic arthritis (JIA). While an addition to scientific literature, I believe that a major revision of the write up is warranted. Please see my comments below:

1- Please keep all percentage points referenced to the same decimal points for consistency. For example, in the introduction, authors write “10.4% in children with polyarticular disease to 41% of patients with the system form.”

This was modified accordingly in the paragraphs as well as in tables.

2- In the introduction, is the main focus on the effects of biologic treatments for growth “retardation” in children with JIA? I think this statement could use a little clarification for the reader.

Indeed, the main focus was to explore the effect of biologic therapy in children with juvenile idiopathic arthritis on growth. Specifically, the question is whether the biologics improve growth in JIA independently of the existence of an initial growth retardation. This clarification was added as requested in the abstract and in the introduction. (Line 77)

3- In the first paragraph of the methods section, can you add more detail to how disputes were resolved? In cases where consensus could not be reached, was a third party consulted to arbitrate the decision?

As requested, we added these clarification were added. “Extracted data from each study was scrutinized independently by both investigators (Dr MY and Dr BAH). They used a pilot-tested extraction form. In cases where consensus could not be reached, a third party (Dr BH) was consulted to resolve disagreements.” (Line 116-Line 121)

4- Instead of stating “The two first authors of this systematic review agreed on the articles to be included in this paper” I would advise to specify the two authors involved as you did in the first paragraph of the methods section.

Thank you for your remark, this was added as requested. (Line 92)

5- For the inclusion criteria, what is meant by “full papers”? Does this mean “full-text papers?” To ensure a robust selection process, please review that studies included were required to meet the following criteria: (1) publication in peer-reviewed journals, (2) relevance to the research question, and (3) availability of sufficient data for extraction in addition to your other already included study specific criteria.

Thank you for your remark. Indeed, we meant full-texts. This was added in the selection criteria section as requested. (Line 97-Line 100)

6- Please also include the date of inception in the first sentence for clarity.

The date of inception was added as requested. (Line 96)

7- In the inclusion criteria section, please review the numbered points on studies selected, they skip number 3.

Thank you for your remark. This was rectified as requested (Line 105)

8- In the inclusion criteria section, this sentence “Studies assessing clinical and/or biological and/or radiological growth parameters before and after biological therapy” could be revised for better clarity.

Thank you for your remark. This sentence was revised to avoid redundancy and enable better comprehension. "Studies evaluating clinical, biological, and radiological growth parameters before and after biological therapy." (Line 104)

9- For number 1 of the exclusion criteria, please add an example of “other autoimmune diseases that may compromise the growth velocity”.

Thank your for remark. An example was added as requested. (Line 108)

10- Please revise the last sentence of the exclusion criteria to improve clarity and maintain a formal tone. Specifically, change "Besides" to "Additionally" and rephrase "not in compliance with the systematic review purpose" to "that did not align with the objectives of the systematic review."

Thank you for your remark. This revision was added to improve clarity for the reader. (Line 113-114)

11- In the Data Extraction and Quality Assessment section, please revise the following sentence “The extracted data included the main methodological characteristics of the articles: study data (year of publication, country, study design, number of included subjects, mean age of included subjects, inclusion and exclusion criteria, duration of the followup)” for better clarity and readability. Specifically, change "included the main methodological characteristics of the articles: study data (year of publication, country, study design, number of included subjects, mean age of included subjects, inclusion and exclusion criteria, duration of the followup)" to "included the main methodological characteristics of the articles, such as the year of publication, country, study design, number of subjects, mean age of subjects, inclusion and exclusion criteria, and duration of follow-up."

Thank you for your remark. This was modified accordingly.(Line 119-121)

12- In the Data Extraction and Quality Assessment section, please fix the spelling of “judgement”

The spelling was corrected as requested. (Line 121)

13- In the Data Extraction and Quality Assessment section, what does “high quality” mean? Please expand on this.

The quality of the studies was identified through Newcastle Ottawa Scale (NOS) for the cohort and case control studies and the revised Cochrane risk-of-bias tool for randomized trials (RoB 2). To avoid confusion for the reader, the sentence has been deleted. The evaluation was the quality of the studies was also pertinent to perfume metaanalysis, which was not the case. Indeed, we could not perform comparison as the methods of evaluation differed among studies.

14- In the results, this statement: “The studies, published between 2003 (17) and 2018 (23),” makes it seem as though there were 17 studies selected in 2003, and 23 studies selected in 2018. Since you state “between”, this is not clear. Please revise for clarity and readability.

The numbers 17,23 referred to first (2003) and last article published (2018). To avoid confusion, the references were added at the end of the sentence. (Line 131)

15- In the results section, “The” does not have to be capitalized before “United States” or “United Kingdom”

Thank you for your remark. This was corrected accordingly. (Line 133)

16- For this statement “Notably, two studies (24,25) were multicenter trials conducted by the members of the Paediatric Rheumatology International Trials Organisation (PRINTO) (24) and the Pediatric Rheumatology Collaborative Study Group (PRCSG) (27): The TENDER trial and the CHERISH trial (28, 29) . The applied designs were the cohort study (13,17–20,23), the open-label non randomized trial (11), the open-label randomized placebo-controlled trial (12,24,25), the case control study (21) and the retrospective study (22)”, I am getting confused on which studies were the “two notable studies.” Please revise the statement for better clarity and to clearly identify the references for the notable studies and study designs. For example, change "Notably, two studies (24,25) were multicenter trials conducted by the members of the Paediatric Rheumatology International Trials Organisation (PRINTO) (24) and the Pediatric Rheumatology Collaborative Study Group (PRCSG) (27): The TENDER trial and the CHERISH trial (28, 29)" to "Notably, two studies, the TENDER trial and the CHERISH trial, were multicenter trials conducted by members of the Paediatric Rheumatology International Trials Organisation (PRINTO) (24) and the Pediatric Rheumatology Collaborative Study Group (PRCSG) (27), respectively."

Thank you for your pertinent remark and for this suggestion. The sentence was reformulated accordingly. (Line 134-137)

17- For the results section, when reporting on patient characteristics, the placing of the references is confusing. Please revise for readability.

Regarding the paragraph of patient characteristics, the references were replaced at the end of the sentences to avoid confusion for the reader. (Line 149-Line 154)

18- Table 1 could be improved for better clarity and readability. Ensure consistent formatting and alignment of columns and rows, and use clear, concise column headers without abbreviations. Separate different types of data into subsections or separate tables, and include footnotes to explain any abbreviations or unique terms. Highlight key data points with bold or italicized text, and add summary statistics like means and medians. Use visual aids to distinguish groups or highlight differences, and maintain consistency in units of measurement throughout the table. Remove redundant information and provide a clear caption summarizing the table’s content along with any relevant notes.

The formatting of the table 1 was revised as requested. We added a column for extremes. We added mean and medians when provided. We maintained the same units and added footnotes when required to facilitate the comprehension of the table.

19- In the results 1.4 Growth Assessment Criteria, this section would read easier in paragraph format for a scientific journal. This follows into sections 1.5 and 1.6, which read more like a paper outline instead of a finished manuscript.

The results were modified as requested. (Line 177-Line 230)

20- Please add a review of all biologics that were included in the studies reviewed prior to the discussion section.

The distribution of the different biologics according to the mechanism of action was recorded in the manuscript as requested. (Line 170-174)

21- The discussion section provides a comprehensive summary of the findings, emphasizing the positive impact of biologic therapies on growth in children with JIA. However, it would benefit from more structured presentation and clearer emphasis on key points.

We organized the discussion into distinct paragraphs with clearer presentation. We also emphasized the most important points specifically the positive impact of biologic therapies on growth.

22- Consider organizing the discussion into distinct paragraphs that separately address the main findings, the implications of these findings, and the limitations of the study. Highlight the most significant improvements in growth parameters, such as HSDS and growth velocity, and discuss their clinical relevance.

We organized the discussion into distinct paragraphs with clearer presentation. We also emphasized the most important points specifically the positive impact of biologic therapies on growth. We also discussed the clinical relevance of such results. (Line 234-291)

23- In the discussion section - while the potential mechanisms and differences based on factors like puberty stage and JIA subtype are well-covered, it would be helpful to further elaborate on the practical implications for treatment strategies and patient management.

Thank you pertinent remark. As requested, we highlighted practical implications on such findings. Specifically, how initiating biologic therapy early in the disease course, is beneficial for patients presenting with growth delays or high disease activity. (Line 241-280)

24- Lastly, explicitly state the limitations of the review, such as heterogeneity in growth assessment criteria and the observational nature of some studies, to provide a balanced perspective.

As requested, the limitations of the review were highlighted. (Line 281-294)

---

## [Decision Letter · Decision Letter 1]

26 Nov 2024

PONE-D-24-30426R1Effect of biologic treatments on growth in children with juvenile idiopathic arthritis: a systematic reviewPLOS ONE

Dear Dr. Makhlouf,

Thank you for submitting your manuscript to PLOS ONE. After careful consideration, we feel that it has merit but does not fully meet PLOS ONE’s publication criteria as it currently stands. Therefore, we invite you to submit a revised version of the manuscript that addresses the points raised during the review process.

The paper can not considered as suitable for publication in present form. Please revise according to the reviewers' comments. 

We look forward to receiving your revised manuscript.

Kind regards,

Martina Ferrillo

Academic Editor

PLOS ONE

Reviewers' comments:

Reviewer's Responses to Questions

**Comments to the Author**

1. If the authors have adequately addressed your comments raised in a previous round of review and you feel that this manuscript is now acceptable for publication, you may indicate that here to bypass the “Comments to the Author” section, enter your conflict of interest statement in the “Confidential to Editor” section, and submit your "Accept" recommendation.

Reviewer #1: All comments have been addressed

2. Is the manuscript technically sound, and do the data support the conclusions?

Reviewer #1: Partly

3. Has the statistical analysis been performed appropriately and rigorously? 

Reviewer #1: I Don't Know

4. Have the authors made all data underlying the findings in their manuscript fully available?

Reviewer #1: Yes

5. Is the manuscript presented in an intelligible fashion and written in standard English?

Reviewer #1: Yes

6. Review Comments to the Author

Reviewer #1: This study is a systematic review examining the effects of biologic treatments on growth in children with juvenile idiopathic arthritis (JIA). The revision notes suggest that editors and previous reviewers have requested numerous adjustments, particularly focusing on methodological consistency, table organization, data collection process, and detailed growth parameters. It appears that several recommendations were made by reviewers, leading to corrections by the authors. The key points and areas of improvement in this article can be summarized as follows:

1. The manuscript contains inconsistencies in percentage expressions and the definition of key concepts such as "growth retardation." The editor has emphasized the importance of clarity and terminological consistency in revisions.

2. The process of data extraction and the resolution of conflicts by including a third party has been improved, but more detailed explanations regarding the assessment of study quality could be beneficial.

3. The studies address various growth parameters; however, the heterogeneity in criteria and the methods used to measure growth outcomes might limit comparability.

4. There are noticeable issues with consistency in tables and figures. The discrepancies between the first table and growth assessment parameters could be resolved through clearer organization and subheadings.

5. While the effects of biologic treatments on growth are thoroughly discussed, practical recommendations for treatment strategies could be more explicitly stated. Especially, potential clinical implications and limitations should be highlighted further.

6. The introduction mentions the impact of biologic treatments on growth retardation, but caution with terms like “retardation” might be necessary. It could clarify whether the focus is exclusively on children with growth retardation or on the general support of growth by biologic treatments. Terminology could be made more explicit.

7. The relationship between biologic treatments, growth mechanisms, chronic inflammation, and the GH–IGF-1 axis is well highlighted. This addition helps readers understand the mechanism of biologic treatments, but referencing foundational studies (e.g., those detailing the effects of chronic inflammation on growth) could further strengthen the introduction.

8. The explanation of how conflicts were resolved during data extraction and review is an improvement. However, specifying the exact criteria used in data extraction and evaluation might be beneficial.

9. A key area noted in the article review is the need for detailed inclusion and exclusion criteria. The term “full papers” has been replaced with “full-text papers,” but a more precise list of criteria for selected articles (e.g., peer-reviewed status, topic-specific data) would enhance the selection process.

10. Table 1 It has been suggested that percentages, age ranges, and other demographic information in the first table be standardized in one format. Presenting different data types with separate subheadings could improve readability.

Adding explanations, captions, and expanded abbreviations to tables would make the data more accessible to readers. Important points (e.g., mean and median statistics) should be highlighted for easier interpretation.

11. The criteria for growth assessment used across different studies are quite varied, making direct comparisons difficult. Standard Deviation Score (SDS) and growth velocity are essential metrics, but additional details on how these metrics were standardized across studies could improve the section.

12. The variability of growth parameters across studies complicates comparisons. For instance, some studies measure IGF-1 levels, while others use SDS or height. Addressing the impact of different measurements on findings could provide valuable context.

13. Although the effects of biologic treatments on growth are broadly summarized in the discussion, it would be helpful to include more explicit information on how these findings could be applied in clinical management. Notably, the potential benefits and clinical outcomes of starting biologic treatment in the early stages of disease could be emphasized.

14. While the discussion acknowledges limitations such as observational design and small sample sizes, further attention could be given to how heterogeneity among studies affects the results and potential solutions to these issues. For example, recommendations for prospective randomized controlled trials to address knowledge gaps in this field could be included.

15. The conclusion summarizes the positive impact of biologic treatments on growth, but it could emphasize the need for further studies on their long-term effects. Comparative analysis of different biologic treatment types, specifically identifying those with the most favorable growth outcomes, would add depth.

16. The importance of integrating growth-promoting strategies with anti-inflammatory treatments in the management of JIA is highlighted. Additionally, specific recommendations for early intervention and regular growth monitoring could enhance clinical applications.

7. PLOS authors have the option to publish the peer review history of their article (what does this mean? ). If published, this will include your full peer review and any attached files.

**Do you want your identity to be public for this peer review?** For information about this choice, including consent withdrawal, please see our Privacy Policy .

Reviewer #1: No

---

## [Author Response · Author response to Decision Letter 2]

11 Feb 2025

Dear reviewers, thank you for your pertinent remarks. We hope that we have adequately responded to all your comments.

Journal comments

Comments to the Author

1. If the authors have adequately addressed your comments raised in a previous round of review and you feel that this manuscript is now acceptable for publication, you may indicate that here to bypass the “Comments to the Author” section, enter your conflict of interest statement in the “Confidential to Editor” section, and submit your "Accept" recommendation.

Reviewer #1: All comments have been addressed

Thank you

2. Is the manuscript technically sound, and do the data support the conclusions?

Reviewer #1: Partly

Indeed, this is not an experiment. All the data were collected from the studies selected for this systematic review.________________________________________

3. Has the statistical analysis been performed appropriately and rigorously?

Reviewer #1: I Don't Know

No statistical analysis was performed for this systematic review. We could not conduct a metanalysis du to the heterogeneity of the studied parameters.

4. Have the authors made all data underlying the findings in their manuscript fully available?

Reviewer #1: Yes

5. Is the manuscript presented in an intelligible fashion and written in standard English?

Reviewer #1: Yes

6. Review Comments to the Author

Reviewer #1: This study is a systematic review examining the effects of biologic treatments on growth in children with juvenile idiopathic arthritis (JIA). The revision notes suggest that editors and previous reviewers have requested numerous adjustments, particularly focusing on methodological consistency, table organization, data collection process, and detailed growth parameters. It appears that several recommendations were made by reviewers, leading to corrections by the authors. The key points and areas of improvement in this article can be summarized as follows:

1. The manuscript contains inconsistencies in percentage expressions and the definition of key concepts such as "growth retardation." The editor has emphasized the importance of clarity and terminological consistency in revisions.

As suggested by the first reviewer, we clarified the definition of “growth retardation” in the context of JIA ensuring it explicitly encompasses a condition of delayed growth based on clinical, biological and radiological data. We have standardized how percentages are expressed across the articles for clearer interpretation. They were expressed as means or adjusted percentages or serum levels

Line 77�77/ 103�106 and Table 2.

2. The process of data extraction and the resolution of conflicts by including a third party has been improved, but more detailed explanations regarding the assessment of study quality could be beneficial.

Thank you for the suggestion. We have added a section discussing the assessment of the criteria used to assess potential biases. We have added this detail in supplementary table risk of biais. (Line 124�126)

3. The studies address various growth parameters; however, the heterogeneity in criteria and the methods used to measure growth outcomes might limit comparability.

Thank you for this pertinent remark. We have acknowledged the heterogeneity in the growth parameters used across the studies. Indeed, the differences in criteria for growth assessment present a limitation when comparing outcomes. This also prevented us from performing metanalysis and was mentioned in the discussion part. (line 64�65 discussion)

4. There are noticeable issues with consistency in tables and figures. The discrepancies between the first table and growth assessment parameters could be resolved through clearer organization and subheadings.

The first table refers to sociodemographic as well as disease characteristics. For Table 1, we standardized how outcomes were expressed. As for the second table, it encompasses growth parameters according to each study. We reorganized the second table which refers to all the variables and outcomes used to express growth parameters. Instead of addressing each author, we reorganized according to the studied parameter. We hope that this version will be more suitable for readers. (Table 1 and Table 2)

5. While the effects of biologic treatments on growth are thoroughly discussed, practical recommendations for treatment strategies could be more explicitly stated. Especially, potential clinical implications and limitations should be highlighted further.

We agree that practical recommendations would enhance the manuscript. In the revised manuscript, we explicitly state that that biologic therapies could be an essential component of the management strategy for growth retardation in children with JIA, offering benefits beyond disease control by potentially enhancing growth. Specifically, initiating biologic therapy early in the disease course, is beneficial for patients presenting with growth delays or high disease activity. (Line 15�Line 20 Discussion)

6. The introduction mentions the impact of biologic treatments on growth retardation, but caution with terms like “retardation” might be necessary. It could clarify whether the focus is exclusively on children with growth retardation or on the general support of growth by biologic treatments. Terminology could be made more explicit.

Thank you for your remark. The term retardation was suggested by the first reviewer. We have clarified the use of the term "growth" to ensure consistency and clarity. We explicitly define "growth " in the context of this review as it focuses on children with JIA who either have initial impaired or normal growth or growth promotion in response to biologic treatments. (Line 104�105)

7. The relationship between biologic treatments, growth mechanisms, chronic inflammation, and the GH–IGF-1 axis is well highlighted. This addition helps readers understand the mechanism of biologic treatments, but referencing foundational studies (e.g., those detailing the effects of chronic inflammation on growth) could further strengthen the introduction.

Thank you for your pertinent remark. We included references to foundational studies that demonstrate how chronic inflammation impacts the growth process. This addition helps readers better understand the biological mechanisms underpinning growth disorders in JIA and the therapeutic potential of biologics. We added in addition to wong et al, two reviews that focuses on this particular aspect. (Line 67 introduction)

8. The explanation of how conflicts were resolved during data extraction and review is an improvement. However, specifying the exact criteria used in data extraction and evaluation might be beneficial.

The criteria used in data extraction were detailed in the method section. We selected original articles (observational, randomized trials) in this systematic review. Studies were included if they assessed: 1) Patients diagnosed with JIA according to the ILAR (International League Against Rheumatism) 2001 classification criteria (1), and treated with biologics. 3) JIA children who who either have initial impaired or normal growth or growth promotion in response to biologic treatments. 3) Studies evaluating clinical, biological, and radiological growth parameters before and after biological therapy. 4) The study was approved by the local Ethics Board and informed consent of parent’s children was obtained. The summary of the process is provided in figure 1. (Line 102�110)

9. A key area noted in the article review is the need for detailed inclusion and exclusion criteria. The term “full papers” has been replaced with “full-text papers,” but a more precise list of criteria for selected articles (e.g., peer-reviewed status, topic-specific data) would enhance the selection process.

As requested, we added the inclusion and exclusion criteria for selected articles: Inclusion criteria: A comprehensive search was conducted from the 1st January 2000 until October 15th 2023. Only full-text papers available and written in English were considered and were required to meet the following criteria: (1) publication in peer-reviewed journals, (2) relevance to the research question, and (3) availability of sufficient data for extraction in addition to the other already included study specific criteria. We selected original articles (observational, randomized trials) in this systematic review. Additional articles were manually retrieved based on the references of selected articles. If any study included overlapping data, the most comprehensive one was selected.

Exclusion criteria: 1) Patients with JIA associated to other autoimmune (juvenile systemic lupus erythematosus) diseases that may compromise the growth velocity, unless they were blended into the studied population and we could not differentiate them when interpreting results. 2) Papers written in another language than English. 3) Publications not representing original research (i.e.; reviews, editorials, qualitative papers, case reports, and letters to editors) were excluded. Additionally, articles that did not align with the objectives of the systematic review were excluded. (Line 95�Line 116)

10. Table 1 It has been suggested that percentages, age ranges, and other demographic information in the first table be standardized in one format. Presenting different data types with separate subheadings could improve readability.

Adding explanations, captions, and expanded abbreviations to tables would make the data more accessible to readers. Important points (e.g., mean and median statistics) should be highlighted for easier interpretation.

We have standardized the presentation of demographic data in Table 1, ensuring that percentages, age ranges, and other relevant information follow a consistent format. We also revised the table to present different data types under separate subheadings to improve clarity and readability. Explanations and captions have been added for better comprehension of the data presented. As requested, we added standard deviation when mentioned, we added captions for age expressed in median instead of mean. We expressed active disease in percentage. (table 1)

11. The criteria for growth assessment used across different studies are quite varied, making direct comparisons difficult. Standard Deviation Score (SDS) and growth velocity are essential metrics, but additional details on how these metrics were standardized across studies could improve the section.

We acknowledge the challenges in comparing these different metrics and suggest that standardization of these measurements in future research would improve the comparability of results. Additionally, we have added more details on how these metrics were defined in different studies and we organized the different parameters used in Table 2.

12. The variability of growth parameters across studies complicates comparisons. For instance, some studies measure IGF-1 levels, while others use SDS or height. Addressing the impact of different measurements on findings could provide valuable context.

The variability in growth parameters across studies indeed complicates direct comparisons, as different metrics such as IGF-1 levels, Standard Deviation Score (SDS), and height capture distinct aspects of growth. To address this, we have added a discussion on how these parameters impact findings. For example, IGF-1 levels primarily capture growth disturbances related to hormonal dysregulation, while SDS and height provide anthropometric measures, making them complementary but not interchangeable. Including such distinctions helps contextualize the findings across studies and emphasizes the importance of selecting growth parameters based on specific study objectives. (Line 11� Line 14, discussion)

13. Although the effects of biologic treatments on growth are broadly summarized in the discussion, it would be helpful to include more explicit information on how these findings could be applied in clinical management. Notably, the potential benefits and clinical outcomes of starting biologic treatment in the early stages of disease could be emphasized.

We have revised the discussion to include more explicit information on the clinical implications of the findings related to biologic treatments and growth. Specifically, we now emphasize how these results can guide clinical decision-making. For instance, we highlight that initiating biologic treatment in the early stages of the disease may provide significant benefits for growth outcomes, as early control of systemic inflammation can mitigate its negative effects on growth velocity and overall development. (Line 15� Line 20, discussion)

14. While the discussion acknowledges limitations such as observational design and small sample sizes, further attention could be given to how heterogeneity among studies affects the results and potential solutions to these issues. For example, recommendations for prospective randomized controlled trials to address knowledge gaps in this field could be included.

We now recommend prospective randomized controlled trials (RCTs) as a key step toward minimizing bias and improving the quality of evidence in this field. Specifically, RCTs could standardize inclusion criteria, define consistent growth assessment parameters (e.g., SDS, growth velocity, or IGF-1 levels), and control for confounding factors such as disease severity and prior treatments. By implementing these methodological improvements, future research could provide more robust and generalizable insights into the effects of biologic treatments on growth. (Line 69�Line 72)

15. The conclusion summarizes the positive impact of biologic treatments on growth, but it could emphasize the need for further studies on their long-term effects. Comparative analysis of different biologic treatment types, specifically identifying those with the most favorable growth outcomes, would add depth.

Thank you for your valuable suggestion. We have revised the conclusion to not only summarize the positive impact of biologic treatments on growth but also to emphasize the critical need for further studies on their long-term effects. In particular, we now highlight the importance of understanding how biologic treatments influence growth over extended periods, as long-term safety and efficacy data are essential for optimizing patient care.Additionally, we have expanded the conclusion to recommend comparative analyses of different biologic treatment types. Identifying which biologics have the most favorable growth outcomes will provide valuable insights for clinicians when making treatment decisions. This comparative ap

---

## [Editor Report · Decision Letter 2]

25 Apr 2025

Effect of biologic treatments on growth in children with juvenile idiopathic arthritis: a systematic review

PONE-D-24-30426R2

Dear Dr. Yasmine Makhlouf,

We’re pleased to inform you that your manuscript has been judged scientifically suitable for publication and will be formally accepted for publication once it meets all outstanding technical requirements.

Kind regards,

Martina Ferrillo

Academic Editor

PLOS ONE

Additional Editor Comments (optional):

Authors modified the text according to the suggestions. I found this work impactful and it fits well with the scope of this journal.

In my opinion, it is suitable for publication in your Journal.

Best regards,

Martina Ferrillo
---

## [Editor Report · Acceptance letter]

PONE-D-24-30426R2

PLOS ONE

Dear Dr. Makhlouf,

I'm pleased to inform you that your manuscript has been deemed suitable for publication in PLOS ONE. Congratulations! Your manuscript is now being handed over to our production team.

Kind regards,

on behalf of

Dr. Martina Ferrillo

Academic Editor

PLOS ONE